# Iterative Dynamic Programming—An Efficient Method for the Validation of Power Flow Control Strategies

**Tom Rüther** [1,2,*] **, Patrick Mößle** [1,2] **, Markus Mühlbauer** [3] **, Oliver Bohlen** [3] **and Michael A. Danzer** [1,2]

1    Chair of Electrical Energy Systems, University of Bayreuth, 95447 Bayreuth, Germany
2    Bavarian Center for Battery Technology, 95447 Bayreuth, Germany
3    Institute for Sustainable Energy Systems, Munich University of Applied Sciences, 80335 Munich, Germany
*    Correspondence: tom.ruether@uni-bayreuth.de

**Abstract:** The operation of electrical networks, microgrids, or heterogeneous battery systems, especially the dispatch of single units within the system, requires sophisticated power flow control strategies. If objectives such as efficiency are demanded for the operation of the energy system, typical control strategies lack the ability to verify the optimality of the operation. Dynamic programming is a widely used method for determining the global optima of trajectory problems. In the context of energy systems and power flow optimization, it is restricted to applications with a low number of states and decisions. The reason for this is the rapid growth of computational effort with increasing dimensionality of the state and decision space. The approach of iterative dynamic programming (iDP) makes dynamic programming applicable to the planning and benchmarking of complex power flow optimization problems. To illustrate this, a heterogeneous battery energy storage system is introduced for which the iDP optimizes the power split at the point of common coupling to minimize the total cumulative loss of energy. The method can be adopted for a broad range of energy systems such as microgrids, utility grids, or electric vehicles. The applicability is limited only by the computation time, which depends on the model complexity and the length of the time series. To verify the functionality of the iterative dynamic programming, its results are directly compared to those of the standard dynamic programming. The total computation time can be reduced by 98% in the tested scenario. As relevant use cases, static and dynamic methods of power sharing are validated and benchmarked. The iDP offers a novel and computationally efficient method for the design and validation of power flow control strategies.

**Keywords:** dynamic programming; multistate systems; computational efficiency; power flow control strategies; heterogeneous battery energy storage system; BESS

## 1. Introduction

Electrical energy systems such as utility grids, microgrids, and battery energy storage systems (BESS) rely on sensible deployment planning and decision making for an optimal utilization of available resources and thus, a minimization of energy losses. A recent report of the International Renewable Energy Agency [1] showed that BESS are an enabling technology for integrating higher shares of renewables and increasing the flexibility of the utility grid. Lithium-ion-based BESS are often part of a microgrid to ensure dispatch and integration of renewable generation and have therefore gained significant importance over the past few years [2,3].

Optimal design and operation strategies are key factors in achieving economic and ecological benefits. In order to ensure these benefits, Li and Wang [4], for example, underlined that achieving multiple objectives, efficient operation, and multilevel collaborative optimization control constitute essential goals for the future development of such systems. According to a study by Bauer et al. [5], these objectives could be either information on operating times and downtimes of the BESS, the overall efficiency of the system, or the impact of

battery degradation on the operating lifetime of the BESS. However, both the configuration (e.g., energy or heterogeneity of the batteries) and the use case (e.g., ancillary services, arbitrage, or peak shaving) of a BESS affect the achievement of these objectives [3,6]. Abedi et al. [7] introduced a comprehensive method for optimal power management and system design. They developed a novel power management strategy integrated into the system dimensioning process of a hybrid energy system with multiple objectives.

For the hierarchical control of microgrids, a distinction is made between three-level and two-level controls [8], wherein this work focuses on the latter. The three-level control is categorized into primary, secondary, and tertiary-level control, differing in the time frame of their respective control objective [8,9]. The main tasks of the control levels can be summarized as follows:

- Primary control (microseconds): Local supervision, voltage and current control, power sharing control.
- Secondary control (milliseconds): Voltage/frequency control restoration, voltage unbalance, harmonic compensation.
- Tertiary control (minutes, hourly): Economic dispatching and optimization.

Typically, the tertiary control is included in the energy management system (EMS), which dispatches one or more energy storage systems [10]. Optimal energy management becomes an essential task in operating a BESS or microgrid beneficially.

Different strategies can be used to optimize the dispatch of EMS, which is strongly dependent on the system and the system boundaries. For hybrid AC/DC microgrids, Hosseinzadeh [11] developed an optimal dispatch to satisfy power demand in both grids solving a mixed-integer linear programming problem. An optimal dispatch for a microgrid in a distribution market environment was introduced by MansourLakouraj et al. [12]. In this way, the microgrid's economic operation, flexibility, and reliability have been improved. This work focuses on power flow control strategies (PFCS) for the EMS in the minute range of tertiary control for heterogeneous BESS according to the work in [13].

For microgrid applications, operational strategies based on state of charge (SoC) balancing have generated considerable research interest [14–16]. For hybrid energy storage systems, research on power and energy management strategies is mainly divided into rule-based control methods such as filter control and optimization-based control methods such as model predictive control [17]. In power system applications, some research in recent years has focused on different optimization techniques, such as model predictive control [18–20], Pareto optimal power flow control [21], or particle swarm optimization [22,23], to achieve multiple objectives. Although all these approaches could improve a BESS's performance, they lack a globally optimal solution.

One way to achieve global optimal solutions is to incorporate a dynamic programming (DP) approach. This allows to evaluate the operation of a BESS and its use cases to set the benchmark for computationally inexpensive PFCSs applied online [24]. In real applications, DP allows the planning of power distribution in EMS ahead of time. Due to the computational effort, the DP is so far only used for BESS problems with a single or a small number of states [25–31]. In [31], for example, a DP approach is utilized to solve an optimal power distribution problem concerning the reduction of the systems' energy losses and the extension of the life cycle of the battery system. However, the problem was simplified by treating the battery's voltage as constant, reducing the complexity of the calculation. In order to overcome such limitations, various algorithm-based improvements have been suggested in the literature.

Kossmann and Stocker [32] introduced an iterative DP approach, substantially different to ours, which optimizes queries to prevent existing memory from overflowing. Here, similar to the greedy algorithm, DP is terminated after a particular criterion. This can be the number of steps, a memory limit, or the computing time. Subsequently, the trajectory with the lowest objective function value is used as the initial value for the further iterations. Another approach based on the greedy algorithm was introduced by Shekita and Young [33], which is also discussed in [32]. However, these approaches are not improved

by recomputing the DP, but rather by sequencing the existing problem. Furthermore, an approximation of DP can be used. Here, the optimal control over a time range is estimated using a neural network. Therefore, the method is called adaptive DP [34]. Huang and Liu [35] used the adaptive DP on residential energy system control. However, the decision space was only differentiated between charge, discharge, and idle. Fuselli et al. [36] extended this approach to the Action-Dependent Heuristic DP. Here, they used two neural networks for home energy resource scheduling. The action network provides the control signal and the critic network evaluates the performance of the action network.

In use cases where a fine discretization is the cause of the computational effort, sequential methods still lead to a high computational time. If neural networks cannot or should not be used due to technical complexity or a lack of reliability, another approach must be chosen to solve the problem. For this reason, the iterative dynamic programming (iDP) is introduced, which iteratively reduces the state and decision space and refines the discretization. For proof of concept, this algorithm is verified by comparing its results to the calculated optima of a standard DP without reduction of the state and decision space. In addition, this algorithm is used to determine the power-sharing factors in an energy system with three BESSs to validate conventional PFCSs.

The main objective of this work is to verify the novel iDP and use it to benchmark existing PFCS. The contributions can be summarized as follows:

- Application of the DP on a multistate, heterogenous BESS for the benchmarking of PFCS;
- Development of the iDP for the efficient computation of the multistate optimization problem;
- Analysis of the computation times and comparison against conventional DP for the metaparameters of the iDP;
- Benchmarking of conventional PFCS through iDP and discussion of the optimal trajectories for different use cases.

## 2. System Description

To study the PFCS without the influence of other generators or loads, the investigated energy system focuses on a heterogeneous BESS. It consists of three parallel lithium-ion batteries with individual DC–DC converters connected to the grid via a central AC–DC inverter through the point of common coupling (PCC). A schematic overview of the system with the used power variables is given in Figure 1. The equivalent circuit diagrams used illustrate the power calculation and are considered in the efficiency maps of the batteries.

### 2.1. Model Design

The batteries, as well as the power electronics, are considered in the applied model. Since the AC–DC inverter is located behind the PCC, as shown in Figure 1, its impact on the power split can be neglected.

The power electronics are represented by efficiency maps. The requested power $P^*$ at the PCC is set as input for the system. The efficiencies of the DC–DC converters are calculated by

$$\eta_{\text{PE},n} = \frac{P_n^*}{P_n^* + c_{0,n} + c_{1,n}P_n^* + c_{2,n}(P_n^*)^2}. \tag{1}$$

Here, the equation for a PV-inverter described in [37] is used. This is admissible because the efficiency curve of an AC–DC converter is similar to that of a DC–DC converter [38]. The adjustment of the efficiency map is achieved via the time-invariant coefficients $c_0$, $c_1$, and $c_2$ and the requested power $P_n^*$ at the $n^{\text{th}}$ battery.

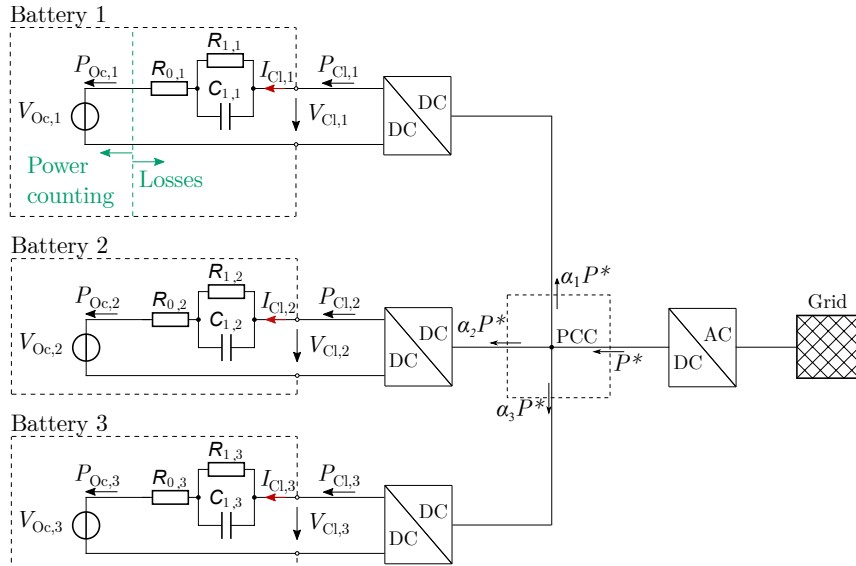

**Figure 1.** Schematic of the BESS with the differentiation of the clamp and open circuit values of the batteries.

An efficiency map based on an equivalent circuit model (ECM) represents the batteries. Thereby, the states of the capacitors no longer have to be displayed in the DP, and the number of states in the system is reduced from six to three, which significantly elevates the computational speed. This effect will be discussed in Section 5.2. Furthermore, the power can be used as an input signal instead of the current, which is more suitable for power flow analysis. However, this leads to a loss of information about the battery clamp- and overvoltages.

The coulombic efficiency, which covers the effects of side reactions [39], is set to 1 and is therefore not considered in this work. Thus, the energy efficiency $\eta_{\mathrm{E}}$ corresponds to the voltage efficiency, which is used for the efficiency maps of the batteries. It is calculated by

$$\eta_{\mathrm{E},n} = \begin{cases} 1 - \dfrac{\int_{t_0}^{\Delta T_\mathrm{e}} P_{\mathrm{Cl},n}(t)dt - \Delta T \cdot P_{\mathrm{Oc},n}}{\int_{t_0}^{\Delta T_\mathrm{e}} P_{\mathrm{Cl},n}(t)dt}, & P_{\mathrm{Cl},n} > 0 \\[3mm] 1 - \dfrac{\Delta T \cdot P_{\mathrm{Oc},n} - \int_{t_0}^{\Delta T_\mathrm{e}} P_{\mathrm{Cl},n}(t)dt}{\Delta T \cdot P_{\mathrm{Oc},n}}, & P_{\mathrm{Cl},n} < 0 \\[3mm] 1, & P_{\mathrm{Cl},n} = 0 \end{cases} \tag{2}$$

and sets the power at the clamps (considering overvoltages) $P_{\mathrm{Cl}}$ into relation with the open circuit power $P_{\mathrm{OC}}$. A specific time range from $t_0$ to $\Delta T_\mathrm{e}$ is investigated, in which a current pulse equivalent to $P_{\mathrm{Cl}}$ is applied to the ECM and both powers are integrated. The duration of the power pulse is chosen such that only minimal changes in state of energy (*SoE*) can be assumed and therefore, the integral of $P_{\mathrm{OC}}$ can be described by multiplying it with the time domain of the integral $\Delta T$. This consequently leads to the representative energy.

The parameters used for the ECM and the used methods for characterization are shown in Figure A1 and Table A1 of the appendix. The energy efficiency is calculated depending on the *SoE* and $P_{\mathrm{Cl}}$, where negative power corresponds to a discharge of the battery. The used efficiency map for the charge and discharge parameters of the ECM and the corresponding values of discretization are shown in Figures A2 and A3. A detailed analysis of the influence of the efficiency maps on the parameters is not provided in this work since the focus is on the iDP approach.

The total energy efficiency $\eta_n$ of the $n$th battery unit can be calculated by

$$\eta_n = \eta_{\text{PE},n} \cdot \eta_{\text{E},n}, \tag{3}$$

taking into account the efficiencies of the power electronics $\eta_{\text{PE}}$ and the battery $\eta_{\text{E}}$. The *SoE* changes over time,

$$SoE_{n,k+1} = SoE_{n,k} + \Delta SoE_{n,k} =$$
$$= SoE_{n,k} + \frac{\Delta T \cdot P_{\text{Bat},n}}{E_{\text{rated},n}} \tag{4}$$

following the time series of the battery power $P_{\text{Bat},n}$. For the discretization of the change of the state over time, a linearized forward Euler method is used. Here, $k$ describes the current time step, $\Delta T$ — the sampling rate, and $E_{\text{rated}}$ — the rated energy of the $n^{\text{th}}$ battery. The actual power charged to/discharged from the $n^{\text{th}}$ battery $P_{\text{Bat,n}}$ takes all mentioned losses into account:

$$P_{\text{Bat},n} = \begin{cases} P_n^* \cdot \eta_n, & P_n^* > 0 \\ \dfrac{P_n^*}{\eta_n}, & P_n^* < 0 \\ 0, & P_n^* = 0 \end{cases} \tag{5}$$

This approach is based on the coulomb counting described in [40], which is applied to powers, instead of currents. Therefore, it is called power counting.

The permitted range of the *SoE* is limited through

$$SoE_{\text{min},n} \leq SoE_n \leq SoE_{\text{max},n} \tag{6}$$

by the minimal value $SoE_{\text{min}}$ and the maximum value $SoE_{\text{max}}$. Those values are set to 0 and 1, respectively. The *SoC* and the *SoE* cannot be directly measured, but can only be estimated by externally measurable parameters. This estimation is affected by the distinct nonlinear behavior of the LIB, considerable changes in battery characteristics over its lifetime due to aging, temperature, voltage noise, etc. For interpretability and clarity of the results, the simple model introduced the Equation (4) is used in this work. A detailed review of different estimation methods is given in [41–43]. Furthermore, the power which can be applied to a single battery is also limited through

$$P_{\text{min},n} \leq P_n^* \leq P_{\text{max},n} \tag{7}$$

by a lower boundary $P_{\text{min}}$ and an upper boundary $P_{\text{max}}$. As a further boundary condition, the requested power of the system must match the sum of the power supplied or received by the battery energy system. This leads to complete fulfillment of the power request by

$$P^* = \sum_{n=1}^{n_{\text{batteries}}} P_n^*. \tag{8}$$

## 2.2. Power Flow Control Strategies

The requested power $P_n^*$ of the $n^{\text{th}}$ battery system is calculated by static, dynamic, and optimized PFCSs through the power-sharing factor $\alpha_n$ and the requested power $P^*$ [3,5]:

$$P_n^* = P^* \cdot \alpha_n, \tag{9}$$
$$n \in [1, 2, 3], \tag{10}$$
$$\alpha \in [0 \ldots 1]. \tag{11}$$

Static PFCSs do not change this factor during operation and rely on the batteries' nominal values or system parameters. On the other hand, dynamic PFCSs adjust the power-sharing factor at every time step and depend on battery states such as the *SoE*. Optimized PFCSs adapt the factor during operation to fulfill a specified objective function.

The static PFCS of **equal power share** considers an equal power split to all battery systems. As shown in

$$\alpha_{\text{num},n} = \frac{1}{n_{\text{batteries}}},\tag{12}$$

it only depends on the total number of batteries $n_{\text{batteries}}$ in the system. [3,5] To adapt the static capacity PFCS method described in [3] to the rated battery energy $E_{\text{rated}}$, the power-sharing factor

$$\alpha_{\text{energy},n} = \frac{E_{\text{rated},n}}{\sum_{i=1}^{n_{\text{batteries}}} E_{\text{rated},i}}\tag{13}$$

is determined. This method is therefore referred to as **rated energy**. Furthermore, the dynamic PFCS method of *SoC* balancing mentioned in [3,16,44–46] can be transferred to the *SoE* **balancing** to calculate the power sharing factor

$$\alpha_{\text{SoE},n,k} = \frac{SoE_{n,k}}{\sum_{i=1}^{n_{\text{batteries}}} SoE_{i,k}}.\tag{14}$$

Therefore, the requested power is shared proportionally to the *SoE* of the $n^{\text{th}}$ battery, concerning the total *SoE* of the system. The calculation is performed again at each time step $k$.

As an optimized PFCS, the iDP approach, described in Section 4, is used to determine a power split considering the global optimum.

### 2.3. Target Indicator

The cumulative loss of energy is used as a target indicator for the evaluation of the PFCSs. This can be expressed through the step-related objective function

$$f_k = \min \sum_{k=1}^{K} \sum_{n=1}^{N} \vec{E}_{\text{L},n,k}.\tag{15}$$

The energy loss of every battery unit $E_{\text{L}}$ is calculated through

$$\vec{E}_{\text{L},n,k} = |\Delta T \vec{P}^*_{n,k}(1 - \vec{\eta}_{n,k})|\tag{16}$$

at every time step and summed up for all $K$ time steps and $N$ Batteries. Since the powers of the batteries are used as decision variables in the DP, the power at each battery becomes a vector that will represent the allowed combinations. The size of this vector depends on the requested power, the state and the discretization of the battery power $P_{\text{discr}}$. However, using a static or dynamic PFCS will degenerate the vector to a scalar because these algorithms produce only a unique solution at every time step.

### 2.4. Load Profiles

The PFCSs mentioned in Section 2.2 are analyzed for an unidirectional and a bidirectional load profile, shown in Figure 2. Both load profiles are normalized to the power $P_{\text{LP}} = 200\,\text{W}$. The magnitude of the load profile can be adjusted without changing the optimization problem itself (provided that the energies are adjusted accordingly), as discussed in Section 5.1. Furthermore, a broad range of the state space and the separation of charging and discharging are considered. The load profile I is based on a peak shaving

profile. The minimum power request (maximum discharge power) is $-0.9$ p.u. and it gradually increases to $-0.15$ p.u. Load profile II consists of a charge and discharge process. Here, a maximum/minimum power of 0.9 p.u. is reached. The power increases/decreases with a rate of 0.15 p.u. per time step. Both profiles have 24 time steps.

The load profiles used here aim at enabling a plausibility check of the iDP in a robust and comprehensible manner. Since the load profiles influence the decision space, they are chosen to differ strongly in their characteristic trajectory. The analysis of highly dynamic load profiles is not presented within this work, as these cannot be adequately checked for plausibility due to the large number of allowed decision trajectories. Furthermore, due to the high computing time, dynamic programming is particularly suitable for applications with high time constants, such as energy management systems. In those, quasi-stationary working points are analyzed in the minute range (or even 15-minute range). Since the time constants of batteries are well below this range, dynamic processes can be neglected. Therefore, less dynamic profiles are better suited for validating the proposed method.

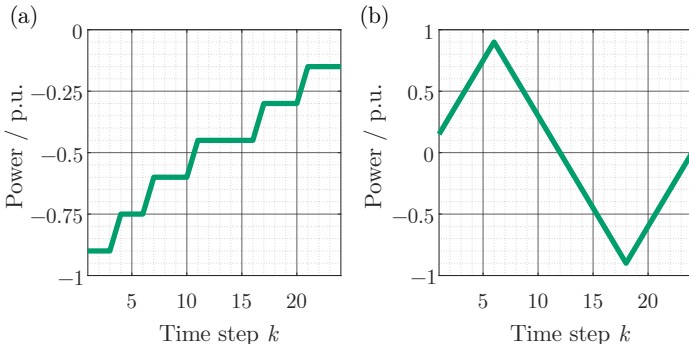

**Figure 2.** Unidirectional load profile I (**a**) and bidirectional load profile II (**b**). Powers are normalized to $P_{\text{LP}}$.

## 3. Dynamic Programming

The DP approach can solve sequential optimization problems, where a global optimum of an overall problem is determined by solving several distributed decisions [47]. The methodology is based on the principle of optimality by Richard Belman, which constitutes an optimal total trajectory as a composition of optimal sub trajectories [47–50].

In general, a minimization of the objective function

$$F(x_1, \ldots, x_n) = \sum_{k=1}^{N} f_k(z_{k-1}, x_k) \tag{17}$$

is performed. The step-related objective function $f_k$ is determined for all time steps until the end of the observation period. This function shows a dependency on the states of the last time step $z_{k-1}$ and the decision of the current time step $x_k$. Therefore, the objective function is not related to any later states of the system, which is a property that must possess all problems solvable with DP [47,49].

Further assumptions and boundary conditions have to be made. The state at time step $k$ has to be accessible by the transfer function $t_k(z_{k-1}, x_k)$. The values of $z_k$ and $x_k$ are confined to the state space $Z_k$ and the decision space $X_k$ respectively. The decision space also depends on the specific state at the current time step. As a non-mandatory condition, the initial state $z_0$ or the final state $z_{\text{end}}$ of the value $\beta_0$ or $\beta_{end}$ can be given. This value is allowed to be a vector. Additionally, both the start and initial set of values can be considered [47,49]. The equation and conditions are summarized as follows:

$$z_k = t_k(z_{k-1}, x_k); \tag{18}$$

$$z_k \in Z_k; \tag{19}$$

$$x_k \in X_k(z_{k-1}); \tag{20}$$

$$z_0 = \beta_0; \tag{21}$$

$$z_{\text{end}} = \beta_{\text{end}}. \tag{22}$$

The DP approach can be performed using either forward or backward recursion. As the computing time depends on the state and decision space sizes, the recursion should be chosen accordingly. If constraints restrict the state space in the initial or final step, the one with the smaller number of states should be selected as starting point. Therefore, the forward recursion should be used if the initial state space is restricted. The same applies vice versa for the backward recursion and is attributed to the slower growth of the state space. This assumption is only valid for a decision space with minor fluctuation, which is considered for the given problem.

For the investigated problem, the objective function (15) is used and the discretization is chosen to be equidistant. The decision space is constrained by Equation (7). However, the *SoE* is used for determining the state space. Since the power counting according to Equation (4) will be used as transfer function, the discretization of the *SoE* can be chosen by taking the minimum power change into account. This corresponds to the discretization of the power $P_{\text{discr}}$. Therefore,

$$SoE_{\text{discr},n} = \frac{P_{\text{discr},n} \min(\vec{\eta}_n)}{E_{\text{rated},n}} \tag{23}$$

is used with the minimum of the efficiency map. The discretization of the *SoE* is calculated separately for each battery system to allow heterogeneity of the parameters. It will only be calculated once and does not change within the time steps. This is permissible to the complete knowledge of the efficiency map according to the discretization and limitation of power. However, the restrictions of Equation (6) are still taken into account.

## 4. Iterative Dynamic Programming Approach

The novel iDP approach constrains the state space iteratively. By doing this, each iteration can refine the discretization at reasonable computing costs. The algorithm does not result in any further restrictions of the decision space, but rather gives an expected state range of the optimal trajectory. The algorithm of the iDP is conducted as follows:

1. Perform a standard DP for the selected load profile as a pre-loop of the iDP with a defined coarse discretization, using the optimal states as input signal for the iDP;
2. Determine the calculation boundaries with a defined bandwidth based on the optimal states of the previous DP (or iDP). The bandwidth size can be chosen as a variable, which is a multiple of the state-space discretization. The bandwidth directly influences the globality and computing time of the solution. A detailed discussion of the influences is given in Section 5;
3. Constrain the time step-dependent state space through the upper and lower values of the calculation boundaries. In terms of batteries, a range of the *SoE* is given;
4. Discretize more finely the constrained, time step-dependent state space. In this work, the discretization of power is chosen to be twice as high for each iteration. According to Equation (23), the discrete state space is also changed;
5. Check whether the termination criterion is met. For example, the number of iteration steps, the change of the value of the objective function, or the calculation time can be used for this purpose. If the criterion is met, the algorithm is stopped and the results are transferred. If this is not the case, it continues with step 6;

6. Perform a DP with the previously defined constraints on the state space. Subsequently, the algorithm jumps back to step 2.

For the example of the termination criterion of a maximum number of iterations $i_{max}$, a schematic program structure is shown in Figure 3a.

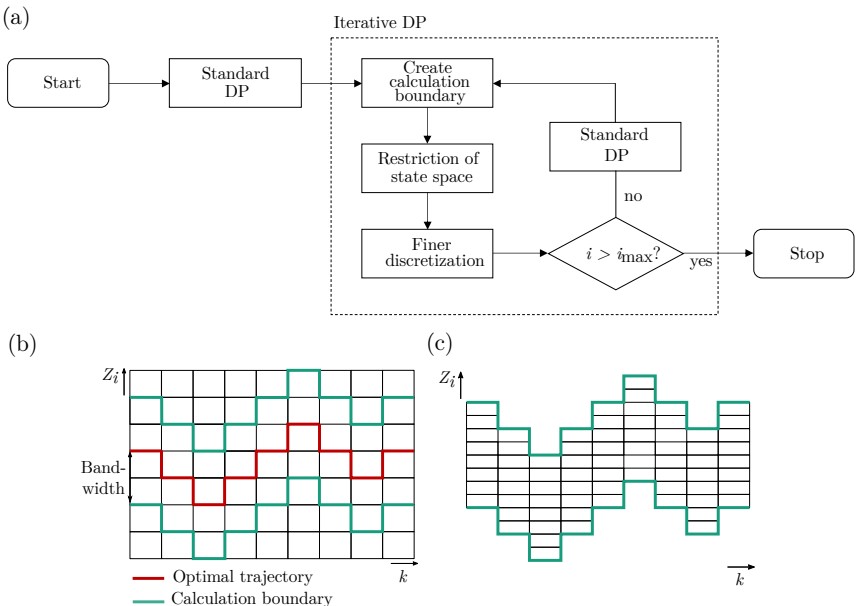

**Figure 3.** Program structure of the iDP approach (**a**), schematic representation of pre-loop with discrete bandwidth of 2 (**b**), and discretization of the first iteration loop (**c**).

Figure 3b illustrates the determination of the bandwidth. Here, a calculation boundary is set around the optimal trajectory of the states of the standard DP of step 1. The bandwidth is a multiple of a discretization step of the standard DP and chosen to be 2 in this example. It is chosen equidistantly in both directions around the optimal state trajectory. The finer discretization from step 4 in the restricted state space is shown in Figure 3c. In both figures, the time step-dependent restriction of the state space is represented.

## 5. Verification and Computing Speed

The iDP is compared with the standard DP to verify the plausibility of the approach. For this purpose, the compliance with the constraints and the comprehensibility of the power split are reviewed. Since the iDP restricts the state space, the solution must also be checked for globality. Furthermore, an investigation in terms of calculation speed is conducted. Here, not only a comparison of the iDP and the DP is made, but also an analysis of the sensitive parameters of the iDP calculation time is drawn. A Windows 10 Enterprise workstation with an Intel Core i5-3570 and 8GB RAM is used as the computing unit. The simulations are performed in Matlab 2019b.

### 5.1. Verification

For the verification, a simplified load profile must be used to calculate the standard DP in reasonable computing time. Therefore, the profile shown in Figure 4a is used for verification. The load profile is only reduced in length and number of possible decisions. It can be easily transferred to a more complex load profile. Due to the reduced number of optimal decision trajectories, the simple decision space provides sufficient comparability with the iDP. The load profile represents an unidirectional load with load levels of 1 p.u., 0.6 p.u., 0.4 p.u., and 0.2. p.u. and a total time range of 15 steps. Considering the difference between time steps $\Delta T$ of 60 s, this corresponds to an overall time of 900 s. The rated energies of all batteries are set to 46.23 W h. Thus, there is no significant change in the *SoE* of the batteries due to the load profile, which simplifies the analysis of the approach.

The starting value of the *SoE* is set to 1, and it is constrained to $0 \leq SoE \leq 1$. The power is limited to the normalized power $P_{\max} = 80\,\text{W}$ for each battery and the discretization of power for the pre-loop is set to 0.25 p.u. As shown in Figure 4c, the constraint of the power boundary is satisfied, even though the power request would allow a higher power for a single battery in the time steps 1 to 8. A more detailed discussion of power and *SoE* constraints is provided in appendix B. The power distribution throughout the BESS is almost equal. This behavior matches the expected outcome for the case given. The minor deviations in the power share between the batteries can be attributed to the coarse discretization. As shown for the first iteration with a power discretization of 0.125 p.u. and a bandwidth of 3, this behavior is reduced. Several trajectories can lead to the same cumulative energy loss. In this study, the objective matrix's first trajectory is always used for analysis. For this reason, the states of the 15th and 1st step do not match the assumption of equal power share. However, since this trajectory leads to the same cumulative energy loss similar to the one of equal power share, it is also a valid solution to the problem of optimality.

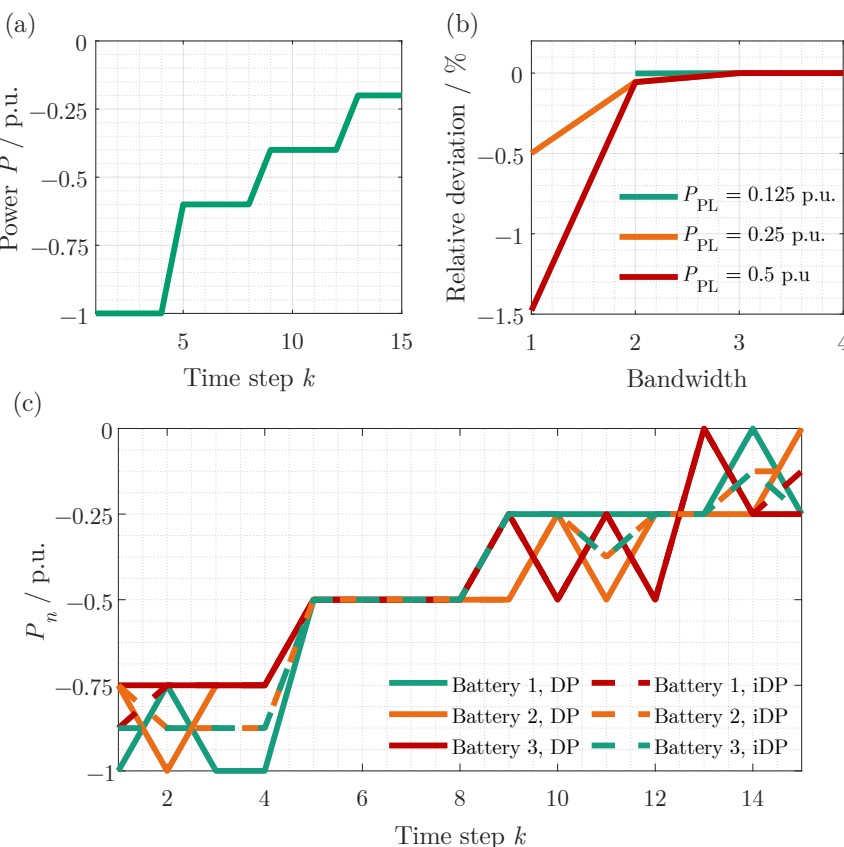

**Figure 4.** Analyzed load profile, normalized to $P_{\text{LP}}$ (**a**), the influence of bandwidth of the first iteration and discretization of the pre-loop on the optimal total loss (**b**), differences in power share of DP and iDP, normalized to $P_{\max}$ (**c**).

In the iDP, the bandwidth and the pre-loop discretization influence the state space, which also effects the solutions globality. Both parameters are analyzed for sensitivity, as shown in Figure 4b. The relative deviation of the cumulative loss of energy of the first iteration to the standard DP is used as a comparative value. For this study, the discretization of the first iteration of the iDP and the discretization of the standard DP are set to 0.0625 p.u. It can be seen that the bandwidth must be chosen to be larger for the coarse discretization to achieve global optimality. With a finer discretization of the pre-loop, the solution is already closer to the global optimum and the state space can therefore be constrained more strictly. Hence, the bandwidth can be chosen to be lower. Since the state space

with coarser discretization is larger for the same bandwidths, it can be concluded that not the size, but the position of the state space has a significant influence on the globality of the solution. According to the discretization of the pre-loop of 0.125 p.u., there is a minimum size of the state space that must be fulfilled due to the non-contiguous state space. The size is determined by its time step dependent restrictions. The discussed effect is increased with high gradients of the load profile that lead to significant changes in the battery states. Therefore, the globality of the solution depends on the two meta-parameters (bandwidth and discretization of pre-loop) of the iDP, which limits its application. In order to choose these parameters in an applicable order of magnitude, preliminary studies must be conducted, as described in this chapter.

Based on Bellman´s theorem, the iDP provides a globally optimal solution within the prescribed solution space. Since this solution space is constrained compared to the original one, other solutions may exist, achieving a lower value of the cumulative objective function. Consequently, the globality of the solution is only given if it is within the solution space. A global solution can be found in the case study by slightly increasing the bandwidth or decreasing the pre-loop discretization. As can be seen in Figure 4b, a convergence to the global optimal solution is given for increasing bandwidths. Even if this global solution is not available within the solution space, only a minor deviation of max. 1.5 % occurs. Therefore, the iDP is considered sufficient for the verification and validation of PFCSs.

By an adaption of the discretization, the iDP is applicable to wide variety of load profiles with other power levels. Thus, there is no significant change in the optimization problem since it is only shifted by magnitude, not complexity. The prerequisite for this is that the order of magnitude also adjusts the energies of the batteries.

### 5.2. Computing Speed

The system and load profiles described in Section 5 are used to investigate the calculation speed. The discretization of power is chosen to be 0.125 p.u. for the standard DP as well as the first iteration of the iDP. Considering a pre-loop discretization of 0.5 p.u. and a bandwidth of 1, the cumulative calculation time for the iDP is 13.31 s. The total calculation time of the DP is 725.16 s, and is accordingly reduced by 98 %. The state-space for the last time step is reduced by a magnitude of 4. The significant influence on computing speed can therefore be attributed to the restricted state space. The size of the state space is furthermore influenced by the decision space, which does affect its growth.

The bandwidth influence on the time step specific calculation speed is shown in Figure 5a for bandwidths from 1 to 10 in a step size of 1. It can be seen that a higher calculation band leads to a longer calculation time. This is attributed to the larger state space and thus, the greater number of computational operations. The effect is also shown in the total and mean value of the calculation time in Table 1. Furthermore, the calculation speed varies with the time step, which is due to the different decision spaces of the time steps. The highest computational effort can be found at the time steps, where each battery energy system can provide the power request itself as well as in combination with other batteries. The standard deviation of the calculation time shows that the influence of the decision combinations is directly coupled with the bandwidth. The minor shifts of the peak characteristics with the same decision space are attributed to the different CPU loads during the simulations. As shown in Figure 5b, the total calculation time of the first iteration depends not only on the bandwidth, but also on the discretization of the pre-loop. Here, the discretization of the first iteration is set to 0.0625 p.u. for all calculation scenarios. The discretization of the pre-loop directly influences the state space since the discretization of the state is directly coupled through Equation (23) to the minimum step of a decision.

Neither the bandwidth, nor the pre-loop discretization has a more significant impact depending on the particular time step, the system, the load profile, and other constraints. For this reason, no statement can be made about the major impact of each parameter for a general case. Therefore, the pre-loop's bandwidth and discretization must be determined empirically. The iDP shows significantly faster computation times than the standard DP.

However, the computational effort remains high and still increases drastically with the number of states considered. Its application is therefore limited to the offline benchmarking of PFCS.

**Table 1.** Computing speed of the iDP for different bandwidths. The mean values and standard deviation are referred per time step.

| Bandwidth | Total Time | Mean Value $\mu$ | Standard Deviation $\sigma$ |
|:---:|:---:|:---:|:---:|
| 1 | 2 s | 0.1 s | 0.1 s |
| 2 | 5.5 s | 0.4 s | 0.3 s |
| 3 | 11.6 s | 0.8 s | 0.4 s |
| 4 | 22.7 s | 1.5 s | 0.9 s |
| 5 | 42.6 s | 2.8 s | 1.8 s |
| 6 | 64.2 s | 4.3 s | 3.1 s |
| 7 | 91.4 s | 6.1 s | 4.8 s |
| 8 | 121.5 s | 8.1 s | 6.6 s |
| 9 | 154.5 s | 10.3 s | 8.8 s |
| 10 | 184.7 s | 12.3 s | 10.8 s |

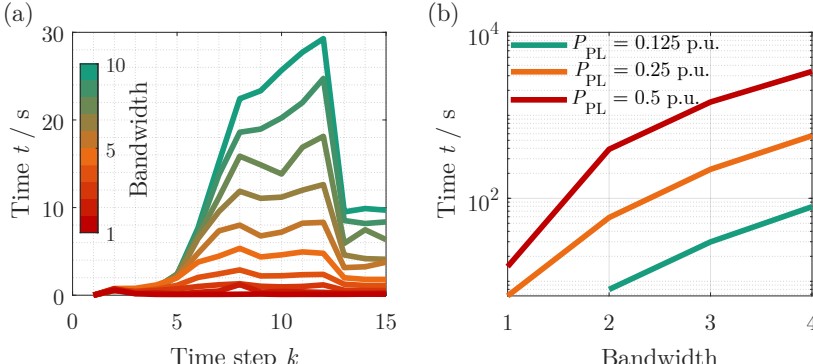

**Figure 5.** Time step-specific computing speed of the iDP on the bandwidth (1 to 10 in a step size of 1) (**a**) and the total computing speed of the first iteration depending on the discretization of the pre-loop as a function of bandwidth (**b**).

## 6. Validation of Power Flow Control Strategies

As the computation time for the iDP remains significantly high, the University of Bayreuth's high-performance computing cluster is used for the calculations in this section. A compute node consisting of two Intel Xeon E5-2630 v4 processors with 20 physical cores and 64 GB RAM is used.

The $\Delta T$ for all simulation scenarios is set to 60 s and the coefficients of the efficiency maps of the converters are set to $c_0 = 0.01$, $c_1 = 0.001$ and $c_2 = 1e^{-6}$ for all power electronics. The power is limited to $P_{max} = 80$ W and $P_{min} = -80$ W for each battery. For simplicity, $P_{max}$ is used to normalize the power of each battery. By varying the load profiles, the influence of the decision space is analyzed. The initial value of the *SoE*, which represents $\beta_0$, is 1 for load profile I and 0.2 for load profile II.

In order to analyze the influence of the state space on the PFCS, the system parameters are varied. Therefore, the energies of the batteries normalized to $E_0 = 30$ W h and are varied according to Table 2. Here, systems I and II describe a homogeneous and systems III and IV a heterogeneous system in terms of energy. The efficiency maps are chosen to be equal to Section 2. Battery I of the systems II and III is reduced by 20 % in energy and efficiency. This represents the decrease in capacity and the increase in internal resistance due to aging, which has been broadly discussed in the literature [51–57]. This also considers the transfer function's influence depending on the different efficiency maps.

**Table 2.** Rated energies of the batteries normalized to $E_0$ in different system configurations.

|           | System I | System II | System III | System IV |
|-----------|----------|-----------|------------|-----------|
| Battery 1 | 1 p.u.   | 0.8 p.u.  | 1 p.u.     | 0.8 p.u.  |
| Battery 2 | 1 p.u.   | 1 p.u.    | 1.25 p.u.  | 1.25 p.u. |
| Battery 3 | 1 p.u.   | 1 p.u.    | 1.5 p.u.   | 1.5 p.u.  |

The power discretization of the pre-loop is set to 0.125 p.u. and iteratively lowered over 0.0625 p.u. to 0.03125 p.u. This results in two iteration stages. The bandwidth is chosen to be 15 for all iterations. The choice was made as a trade-off between available computing time on the cluster and discretization of the solution.

*6.1. Results*

The cumulative energy losses for all systems with different PFCSs, taking load profile 1 into account, are shown in Table 3. The energy losses are given relative to the total energy of the system. It can be seen that the iDP obtains the lowest cumulative loss of energy in all analyzed systems. All other PFCSs have the same cumulative energy loss, which differs from the iDP by 0.17 %. System II achieved the greatest loss of energy for the PFCS of equal power share, followed by the methods of rated energy and *SoE* balancing. The conventional methods have a maximum deviation of 11.99 % from each other, while there is a minimal deviation of the *SoE* balancing from the iDP of 137.6 %.

In system III the PFCSs of *SoE* balancing and equal power share achieve almost an equal loss of energy. The method of rated energy has the highest value and thus deviates at least by 2.45 % from the other methods. The deviation of the PFCS of equal power share differs from the iDP by 0.32 %. In System IV, the PFCS of equal power share, analogous to system II, has the greatest energy loss, followed by the PFCS of *SoE* balancing and rated energy. The minimum deviation to the iDP is 106.51 %.

**Table 3.** Cumulative loss of energy of different power split methods for load profile I. Energy losses are given relative to the total energy of the system.

| PFCS             | System I | System II | System III | System IV |
|------------------|----------|-----------|------------|-----------|
| Equal power share| 9.35 ‰   | 42.50 ‰   | 7.49 ‰     | 32.6 ‰    |
| Rated energy     | 9.35 ‰   | 36.89 ‰   | 7.68 ‰     | 24.94 ‰   |
| SoE balancing    | 9.35 ‰   | 38.22 ‰   | 7.5 ‰      | 29.25 ‰   |
| iDP              | 9.33 ‰   | 15.52 ‰   | 7.47 ‰     | 12.07 ‰   |

For load profile II (Table 4), the methods have the same distribution of the energy losses as for load profile I. However, there are different quantities of energy. The minimum deviations of the best method to iDP are 0.28 %, 127 %, 0.38 %, and 94.34 % for systems I, II, III, and IV. For all calculated scenarios, the iDP has the lowest cumulative loss energy. The power distribution of all PFCS for the different load profile–system combinations are shown in the supplementary materials.

**Table 4.** Cumulative loss of energy of different power split methods for load profile II. Energy losses are given relative to the total energy of the system.

| PFCS             | System I | System II | System III | System IV |
|------------------|----------|-----------|------------|-----------|
| Equal power share| 10 ‰     | 39.24 ‰   | 8.07 ‰     | 31.02 ‰   |
| Rated energy     | 10 ‰     | 35.22 ‰   | 8.23 ‰     | 24.11 ‰   |
| SoE balancing    | 10 ‰     | 37.6 ‰    | 8.07 ‰     | 31.02 ‰   |
| iDP              | 9.97 ‰   | 15.51 ‰   | 8.04 ‰     | 12.41 ‰   |

*6.2. Discussion*

In system I, all PFCSs without the iDP degenerate to the method of equal power share, due to the complete homogenization of the system. The power is shared equally in every time step. Since the iDP also considers the losses of the DC–DC converters, no equal share of the power is drawn out for this control strategy in time steps 21 to 24. This is attributed to the low efficiency of power electronics in low-power areas. The deviation of the other PFCSs to the iDP can be neglected due to the small difference.

The PFCS of equal power share yields the highest loss of energy for system II since battery 1 got an equal power share although its efficiency is decreased. The method of rated energy as well as the *SoE* balancing include more information about the system's parameters and therefore shows a higher performance. The PFCS of rated energy achieves a lower value of cumulative loss of energy. This is due to the reduced power of battery 1 already starting at the beginning. The *SoE* balancing slightly reduces the power share of battery 1 over time due to its relative reduction of the *SoE*. However, the remaining deviation from the iDP is significantly high. This corresponds to the fact that the iDP uses battery 1 only in the time steps 1 to 3 in which it is used to fulfill the requested power boundary condition. The other two batteries cannot fulfill the condition due to the power limitation of 1 p.u.

Due to the different battery energies in system III, there are deviating energy losses between the different PFCSs. The PFCS of rated energy has the highest cumulative energy loss since all batteries are reduced to the same *SoE* and, therefore, the same efficiency. In the method of *SoE* balancing and equal power share, the *SoE*s are distributed at different levels due to the heterogeneous energy distribution. Batteries with higher energy are kept in areas of higher *SoE* and thus also in an area of higher efficiency, as can be seen in Figures A2 and A3. The minimum deviation to the iDP is attributed to the optimal power distribution in low-efficiency areas of the power electronics. In System IV, the PFCS of equal power share, analogous to system II, has the highest energy loss due to the equal power share in a system with different efficiencies. The same relationships apply to System II for *SoE* balancing and the rated energy method. However, due to the heterogeneous energy distribution of the system, the differences become more pronounced in the distribution of the loss energies. For the iDP, analogous to system II, battery 1 is only used in the iDP to fulfill the power request.

For load profile II, system I behaves equally to load profile I. Again, for low-power requests, only two batteries are used, to achieve maximum overall efficiency in terms of iDP.

In System II, the method of rated energy remains with the least deviation to the iDP. The *SoE* balancing results are used in a superimposed effect of reduced efficiency and energy. This leads to an equal increase in the *SoE*s of all batteries in the charging case. At discharge, the *SoE* of battery 1 drops significantly faster than that of the other batteries and is therefore discharged with lower power. In total, battery 1 is used less in terms of total power with the *SoE* balancing than the control strategy of equal power share. This is also underlined by the behavior of the iDP, which only uses battery 1 to meet peak loads.

Systems III and IV behave similarly in load profiles I and II in terms of conventional PFCSs. The iDP makes the same decisions in system III as in system I, for load profile II. The reason for this is also the superposition of the efficiency maps of the DC–DC converters and the batteries. In system IV, battery 1 is also used only by the iDP to fulfill the power request.

In general, the procedures of static and dynamic PFCSs show a strong dependency on the state space (system configuration) and transfer function (reduced efficiency map). An influence of the performance ranking on the load profile could not be determined. However, for systems I and III all PFCSs show good results close to the optimum found by the iDP. The decrease in the efficiency of battery 1 leads to a strong deviation of the procedures from the optimal solution. This is particularly evident in the significantly different power distributions of the iDP. Due to the varying performance for the evaluated system configurations, no conventional PFCS can be used across the board. Therefore, it

is recommended to verify and validate different real-time capable methods by iDP before starting the application.

## 7. Conclusions

In this study, an iterative dynamic programming approach is introduced and validated based on a multistate BESS. The system is described by efficiency maps. System variables are transferred to the state and decision space in order to make them applicable for DP.

Since the DP has a high computing time, which increases significantly with the number of states and the discretization, the state space is restricted using the iDP approach. For this purpose, the DP is initially executed in a pre-loop with coarse discretization to estimate the optimal states. Subsequently, a calculation band is placed around the optimal states and the discretization is iteratively increased. The approach is validated against the conventional DP and it can be stated that the global optimum can be found within a sufficient bandwidth. Since there is currently no concrete predictive method for determining the bandwidth, it must be assumed that only an optimum close to the global one can be found in real applications. However, the analysis of a test profile has shown a reduction in computing time of 98.16 %, making the iDP applicable to multistate problems.

Verification and validation of existing dynamic and static PFCSs in terms of cumulative loss of energy are possible application examples of the iDP. Using this, not only an evaluation of the procedures among each other, but also the analysis of the deviation from an optimum is possible. It can be shown that influencing the transfer function by a reduction of the efficiency strongly varies the performance of the conventional PFCSs. There are deviations of up to 137.6 % of the cumulative loss of energy compared to the iDP. For systems with only a change in state space, the conventional PFCSs have a maximum deviation of 0.38 %. Therefore, especially for the former, significant savings in operating costs can be achieved.

The iDP method can be used to design, develop, and validate PFCSs. In contrast to dynamic PFCSs, the iDP approach does not revert to predefined dependencies on state or operation variables. The analysis of the results of the near-optimal solutions of the iDP thus allows to identify dependencies that have until now been disregarded. With that, dynamic PFCSs can be improved. In addition, it can also be used to evaluate optimization-based approaches that do not guarantee globality in their solution, such as model predictive control.

Since the iDP is limited by the metaparameters of bandwidth and discretization of the pre-loop, further development must be made. For example, an adaptive grid for the bandwidth could be developed, which gets adjusted based on the gradient of the optimal decision. Furthermore, the development of a predictive method for determining the necessary state band for a solution with global optimum criteria will be addressed in future work. Another limitation of the method remains through the still high computational effort compared to other PFCS. Initial studies have shown that a proper choice of metaparameters can tackle this. Nevertheless, improving the algorithm in terms of the computational effort is still necessary to make it applicable to a broader range of problems.

In future work, the model of the BESS will be extended by an aging model to enable the optimization of both operating and investment costs and thus reduce the levelized cost of electricity (LCOE) and levelized cost of storage (LCOS), respectively. In addition to energy efficiency and system economics, ecological targets can be taken into account within a multifactorial evaluation.

**Author Contributions:** Conceptualization, T.R. and P.M.; methodology, T.R., P.M., M.M. and M.A.D.; software, T.R.; validation, T.R.; formal analysis, T.R.; investigation, T.R.; data curation, T.R.; writing—original draft preparation, T.R., P.M., and M.M.; writing—review and editing, T.R., P.M., M.M., O.B., M.A.D.; visualization, T.R.; supervision, O.B., M.A.D. All authors have read and agreed to the published version of the manuscript

**Funding:** Funded by the Deutsche Forschungsgemeinschaft (DFG, German Research Foundation)—491183248. Funded by the Open Access Publishing Fund of the University of Bayreuth.

**Institutional Review Board Statement:** Not applicable.

**Informed Consent Statement:** Not applicable.

**Data Availability Statement:** Data available on request.

**Acknowledgments:** The authors would like to thank the Bayreuther Center for High Performance Computing (BZHPC) of the research center for scientific computing at the University of Bayreuth for providing access to the high-performance computing cluster.

**Conflicts of Interest:** The authors declare no conflict of interest.

**Abbreviations**

The following abbreviations are used in this manuscript:

| | |
|---|---|
| BESS | Battery energy storage system |
| DP | Dynamic programming |
| ECM | Equivalent circuit model |
| EMS | Energy management system |
| iDP | Iterative dynamic programming |
| LCOE | Levelized cost of electricity |
| LCOS | Levelized cost of storage |
| PCC | Point of common coupling |
| PFCS | Power flow control strategies |
| SoC | State of charge |
| SoE | State of energy |

**Appendix A. Paramters and Efficiency Maps**

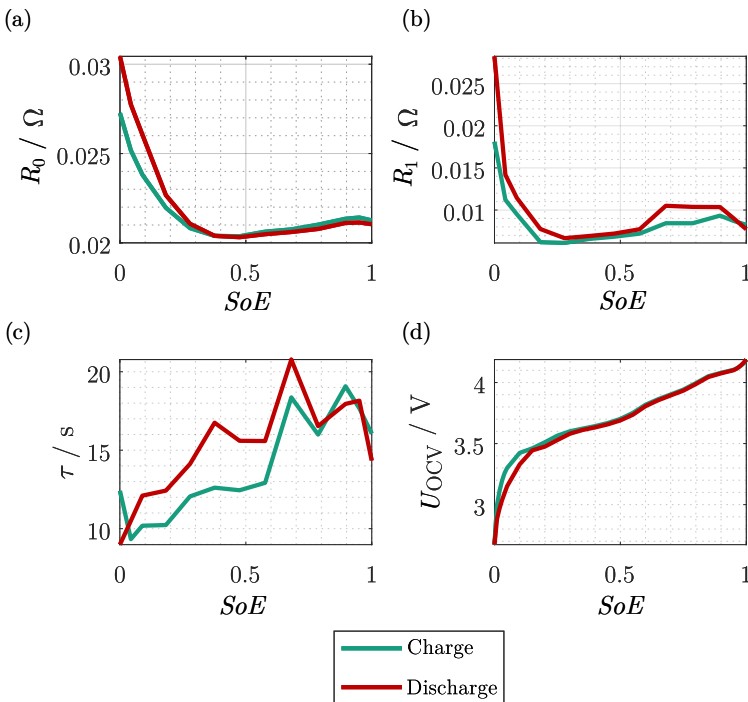

**Figure A1.** Chosen Parameters of the ECM for an NCA 18650 cell. Electrolyte and line resistance (**a**), charge transfer resistance (**b**), time constant of the charge transfer (**c**), and open circuit voltage (**d**).

**Table A1.** Parameter determination of the ECM.

| Parameter | Determination |
|---|---|
| Open circuit voltage | Incremental method at 25 °C inside a climate chamber Puls test with a current of 2C and a pulse time of 10 s. |
| ECM Parameters | Determination of the parameters by the voltage relaxation and a lsqnonlin fit to the ECM. |

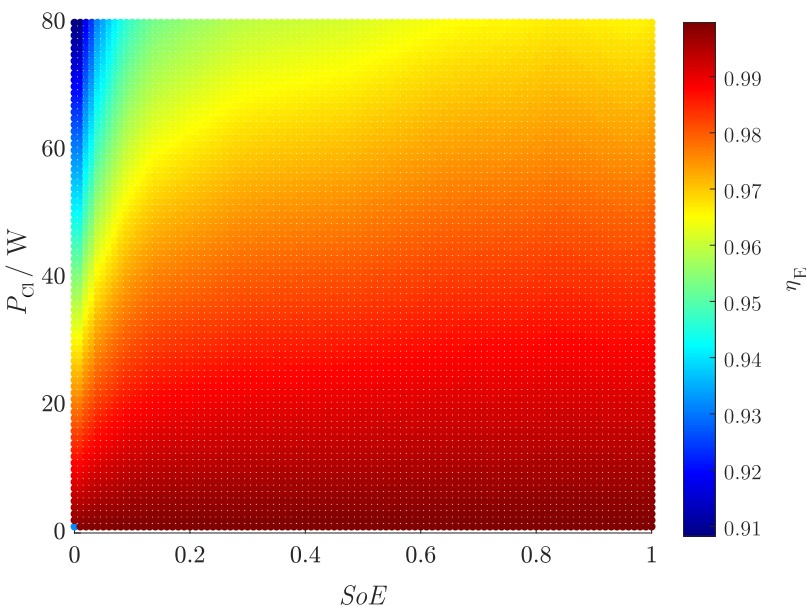

**Figure A2.** Efficiency map considering the charge parameters with a discretization of the power of 1 W and a discretization of the *SoE* of 1%.

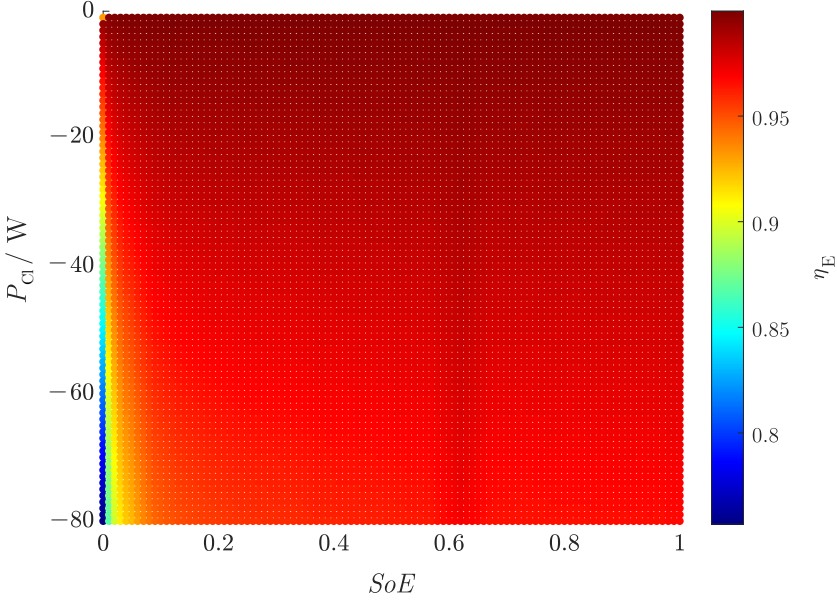

**Figure A3.** Efficiency map considering the discharge parameters with a discretization of the power of 1 W and a discretization of the *SoE* of 1%.

## Appendix B. Boundary Value Test

A boundary value test is performed to check compliance with the constraints of the states and decisions. The performance limits are set to $-30\,\text{W} \leq P \leq 30\,\text{W}$ and the SoE

limits are defined to $0 \leq SoE \leq 1$. The BESS is loaded with a constant power of 30 W for a time horizon of 40 steps (starting with step 1). A charging and discharging case are examined. In both cases, the *SoE* is set to 0.5 at the beginning. All efficiencies are set to 1 to simplify the analysis. The homogeneous system consist of three batteries which have an energy of 0.22 W h to enable complete charge/discharge of the battery in a low number of time steps. For the sake of of simplicity, the difference between the time steps is chosen to be 1 s. The optimal decision and state trajectories are shown in Figure A4.

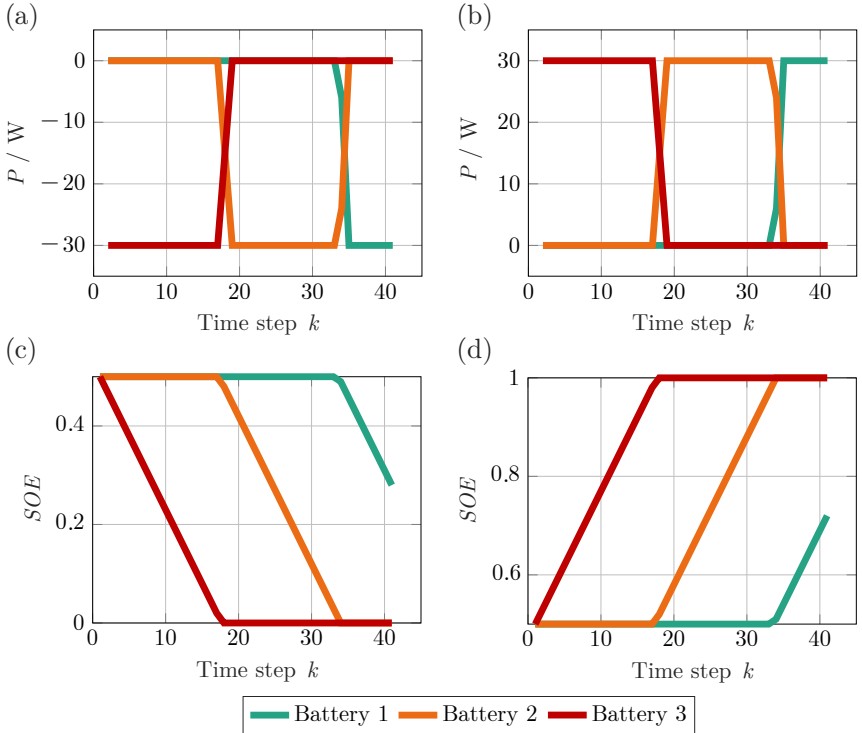

**Figure A4.** Optimal decision variables for the discharging (**a**) and charging (**b**) cases, as well as optimal states of the discharging- (**c**) and charging (**d**) cases of the boundary value verification.

The batteries are charged and discharged after the other up to the *SoE* limit of 1 and 0, respectively. Once one battery reaches the corresponding limit, the next battery provides the power request. Before reaching this limit, the power is divided between the battery currently providing power and the battery that will provide power in the future since the total power demand is no longer available from the current battery. The sequence of the requested batteries is due to the loop order of the decision variable function.The power limit is complied with in the cases considered. However, the maximum power demand corresponds to that of the batteries and is not exceeded. In Section 5, the power limitation is therefore checked by a load profile with a higher amplitude.

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
