# Peer review of "Iterative Dynamic Programming—An Efficient Method for the Validation of Power Flow Control Strategies"

_electricity, doi:10.3390/electricity3040027_

Round 1

Reviewer 1 Report

In this paper the authors study the methods for the validation of power flow control strategies. Consistent power system operation is very important to both power utilities and consumers. This paper proposes power flow control strategies (PFCS) for the energy management systems (EMS) in the minute range of tertiary control for heterogeneous battery energy storage systems (BESS).
The topic is quite interesting, but I have comments.
The approach proposed in this article PFCS is interesting, but not very effective in practice. This is a beautiful theoretical mathematical model. The authors of the article need to show the applicability of the proposed model using a specific example of an electrical network. For this, it is necessary to take an electric network, for example, 110 kV or 10 kV, and consider various options EMS. Then readers will understand the importance of your research, which is presented in the article.
In my opinion, in the article it is necessary to provide a drawing of the electrical network and provide several options for graphs of electrical loads. For example, consider the electrical load schedule for household and industrial consumers. At the same time, the BESS discharge rate will be significantly different for both cases.
Also, in the article, it was possible to perform a forecast of the electrical energy consumption schedule using a neural network. And then use this graph to simulate the operation of BESS.
The presented report is at a very high scientific level. I believe that the present study has a significant scientific and applied contribution, which is strongly emphasized in the basically reporting volume. A slight clarification can be made in the abstract part, where the quality of the research can be enhanced. In the conclusions, it is necessary to describe what economic effect can be obtained using this method? How much can the service life of BESS increase?

Reviewer 2 Report

This manuscript proposes iterative dynamic programming (iDP) to achieve optimal global solutions for the power split and minimize the total cumulative energy loss for a heterogeneous battery energy storage system.

The manuscript is very well written and presented.

Although the iDP is applied only in a conceptual model, the results are a start point for more research in this area.

Just some minors:

- The authors should make clear the limitations of this work.

- Line 232 - Latex \section{}

Reviewer 3 Report

Comments to the authors

Manuscript ID: electricity-1857118

Title: Iterative dynamic programming-An efficient method for the validation of power flow control strategies

1) The first paragraph of the introduction is a bit misleading; it emphasizes battery storage systems.

2) Literature review is very poor. Many relevant publications are skipped. For instance, [R1]-[R3] also provide an optimization-based power management scheme that could be discussed in the submitted manuscript:

[R1]. “A comprehensive method for optimal power management and design of hybrid RES-based autonomous energy systems”, 2021. [https://doi.org/10.1016/j.rser.2011.11.030]

[R2]. “Robust Optimal Power Management System for a Hybrid AC/DC Micro-Grid”, 2015. [https://doi.org/10.1109/TSTE.2015.2405935]

[R3]. “Optimal power management of dependent microgrid considering distribution market and unused power capacity”, 2020. [https://doi.org/10.1016/j.energy.2020.117551]

3) Provide the source of all equations if not driven by yourselves. For instance, what is the source of Equation (1)?

4) Calculating the state of charge for batteries are not easy. It is recommended to comment on that just after Equation (6).

5) This manuscripts lacks a proper comparison study with baseline methods. For instance, the authors need to compare their method with traditional dynamic programming.  

Reviewer 4 Report

This paper deals with the iterative dynamic programming for power flow strategies. The subject is very interesting and the respective results of the applied method are very satisfactory. There are few points, which should be explained. More specifically:

è There are some parameters, which are not explained or explained after mediation of main lines. E.g. in line 123 “SoE” is presented, but in line 128 it is explained. “ηPE” in eq. (3) and “P*n” in eq. (5) have not been explained directly.

è In line 232 something is missing in phrase “sectionVerification and computing speed”.

è A minus typographical error in line 442 (“… a changes…” and in line 490 (“…chapter…” instead of  “… section…”)

Round 2

Reviewer 3 Report

No further comment.